# Effects of Altering Magnesium Metal Surfaces on Degradation In Vitro and In Vivo during Peripheral Nerve Regeneration

**DOI:** 10.3390/ma16031195

**Published:** 2023-01-30

**Authors:** Rigwed Tatu, Leon G. White, Yeoheung Yun, Tracy Hopkins, Xiaoxian An, Ahmed Ashraf, Kevin J. Little, Meir Hershcovitch, David B. Hom, Sarah Pixley

**Affiliations:** 1Department of Pharmacology & Systems Physiology, College of Medicine, University of Cincinnati, Cincinnati, OH 45267, USA; 2PSN Labs (Plastics Services Network), Erie, PA 16510, USA; 3Department of Mechanical Engineering, Department of Bioengineering, North Carolina Agricultural & Technical State University, Greensboro, NC 27411, USA; 4Northrop Grumman, Baltimore, MD 21240, USA; 5Division of Pharmaceutical Sciences, James L. Winkle College of Pharmacy, University of Cincinnati, Cincinnati, OH 45229, USA; 6College of Medicine, University of Cincinnati, Cincinnati, OH 45267, USA; 7Head & Neck Surgery, San Diego Medical Center, University of California, San Diego, CA 92103, USA

**Keywords:** magnesium, PEO anodization, in vitro vs. in vivo, peripheral nerve regeneration, bioabsorbable metal, micro-CT

## Abstract

In vivo use of biodegradable magnesium (Mg) metal can be plagued by too rapid a degradation rate that removes metal support before physiological function is repaired. To advance the use of Mg biomedical implants, the degradation rate may need to be adjusted. We previously demonstrated that pure Mg filaments used in a nerve repair scaffold were compatible with regenerating peripheral nerve tissues, reduced inflammation, and improved axonal numbers across a short—but not long—gap in sciatic nerves in rats. To determine if the repair of longer gaps would be improved by a slower Mg degradation rate, we tested, in vitro and in vivo, the effects of Mg filament polishing followed by anodization using plasma electrolytic oxidation (PEO) with non-toxic electrolytes. Polishing removed oxidation products from the surface of as-received (unpolished) filaments, exposed more Mg on the surface, produced a smoother surface, slowed in vitro Mg degradation over four weeks after immersion in a physiological solution, and improved attachment of cultured epithelial cells. In vivo, treated Mg filaments were used to repair longer (15 mm) injury gaps in adult rat sciatic nerves after placement inside hollow poly (caprolactone) nerve conduits. The addition of single Mg or control titanium filaments was compared to empty conduits (negative control) and isografts (nerves from donor rats, positive control). After six weeks in vivo, live animal imaging with micro computed tomography (micro-CT) showed that Mg metal degradation rates were slowed by polishing vs. as-received Mg, but not by anodization, which introduced greater variability. After 14 weeks in vivo, functional return was seen only with isograft controls. However, within Mg filament groups, the amount of axonal growth across the injury site was improved with slower Mg degradation rates. Thus, anodization slowed degradation in vitro but not in vivo, and degradation rates do affect nerve regeneration.

## 1. Introduction

Biodegradable medical implants are desirable in many tissue repair situations because the materials degrade within the body safely and completely. Mg metal is especially interesting as a biomedical implant material because it is an essential mineral found in the body and Mg ions are crucial for the function of hundreds of proteins and thousands of cellular functions. Mg metal was first tested for use as a medical implant around 1900, but problems with production and impurities delayed significant interest [1]. With improved manufacturing processes, the interest in the exciting possibilities of using Mg metal has risen exponentially since WWII. The major uses explored with Mg metal have been for orthopedic and vascular stent applications. However, we have extended work with Mg metal to a soft tissue repair application: peripheral nerve regeneration after injury.

Peripheral nerve injury (PNI) affects millions of people worldwide due to both the initial injury and the resulting disabilities and loss of productivity that can have long-lasting effects. PNI due to traumatic injury in the civilian population occurs in ~2–5% of all emergency room visits in the U.S. and surgeries to repair PNI damage have been estimated at ~560,000 per year in the U.S., and this has resulted in an estimate of the market for new and improved methods of nerve repair of $1.3–1.9 billion [2,3].

PNI that results in complete severance of a nerve requires surgery to reconnect the cut ends of the nerves, especially if sufficient nerve segments are lost. The distal connective tissue wrappings support and guide regenerating axons back to their original site. Direct reconnection is best, but if the gap is ~9 mm or longer, then the tension created can cause further damage [3]. For longer gaps, a substitute for the lost nerve tissue, a scaffold, is sought and this should duplicate the normal, complex structure of a PN. The current clinical “gold standard” of scaffolds is an autograft, a nerve taken from the same patient. However, this involves additional patient damage and a second surgical site and may still not achieve total return of function or sensation [4]. Significant research has identified hollow nerve guidance conduits made of biomaterials for clinical repairs and several are in clinical use, but they are only effective for what are termed non-critical (shorter) nerve gaps of <20 mm in humans [3,5]. In non-clinical rodent experiments, the critical gap size past which hollow nerve conduits are low to ineffective at repair is 14–15 mm [6]. Therefore, the current clinical and research challenge is to improve biomaterial conduits to repair critical-length nerve injury gaps.

Preclinical research has identified several factors that can improve the success of hollow nerve conduits for repair of both short and longer gaps [7,8,9]. One factor determined to improve repair is to provide linear materials inside hollow nerve conduits. Linear materials, such as strands or small channels, when placed within hollow nerve conduits, duplicate the normal internal connective tissue structure of a nerve. Linear materials also mimic the situation in gaps below the critical limit where proteins from clot materials condense to form fibrin “cables” that serve this purpose; the cables do not form over longer distances [6].

To our knowledge, we are the only lab that has proposed and studied the use of biodegradable Mg in the form of thin filaments (<400 µm diameter) to serve a contact guidance function for nerve repair. To date, we have demonstrated that scaffolds consisting of single Mg filaments inside hollow nerve conduits were biocompatible with regenerating nerve tissues, that Mg degradation did not leave scars, and that Mg use reduced inflammation in the regenerated tissues [10,11]. We also demonstrated that Mg filaments increased axonal content within fascicles, but only for non-critical nerve gaps of 6 mm, not for a critical gap of 15 mm, nor was function improved with the longer gap [11]. Towards understanding how to improve the use of Mg to support nerve regeneration across longer gaps in rats, we speculated that the Mg filaments inside the longer gap may have degraded too rapidly and did not remain intact long enough to provide adequate support. This was based on our previous experiments, where we observed gaps in the Mg filaments by six weeks after surgery [10,11]. In addition, a slower degradation rate will eventually be necessary for use in humans, where the gap size could be up to 130 mm [3]. Rapid degradation of this very reactive metal is also known to break down water to produce hydrogen gas bubbles and OH^−^ radicals [12]. While the increased local pH appears to be well buffered in vivo, the gas bubbles can press on local tissues and cause concerns. We observed some evidence of bubbling adjacent to the Mg, inside a conduit, in at least one animal after six weeks [11]. We speculated that Mg metal degradation and bubbling due to contact with fluids would be more significant in a longer gap, which might have resulted in a too rapid degradation and interfered with neuronal cell attachment or outgrowth. Therefore, this study examined methods to slow the degradation rate of Mg metal in vivo.

Many methods are used to reduce the corrosion rate of soft metals, and many have been tested with Mg [13,14]. Of these, microarc oxidation (MAO), which is also known as plasma electrolytic oxidation (PEO), is one of the most promising, as this produces a protective oxide layer that slows the rate of corrosion attack during the initial implantation period and therefore would reduce hydrogen evolution and the rise in pH [14]. PEO is a process commonly used to protect soft metals and involves a high-voltage plasma-assisted anodic oxidation where the plasma discharges created during the process cause partial short-term melting of the oxide layer and lead to the formation of a highly adherent ceramic oxide coating [15,16]. PEO is a relatively inexpensive and environmentally friendly technique, especially when non-toxic elements are used. The resulting coatings from the PEO method possess sufficient wear and corrosion resistance as well as desirable electrical properties and high thermal stability [15,16,17,18,19,20].

Here, we report on how pure (99.9%) Mg filaments of 250 µm diameter were prepared by polishing and PEO anodization with non-toxic materials, and then tested both in vitro and in vivo for degradation, biocompatibility, and effectiveness in nerve repair. The polishing treatment of as-received Mg filaments exposed more Mg at the surface. Subsequent PEO treatment slowed degradation of Mg metal filaments in vitro, after four weeks of immersion in a physiological salt solution, and improved attachment of cultured cells. When used inside a hollow nerve conduit to repair a critical-length (15 mm) gap in rodent sciatic nerves, all forms of Mg were compatible with nerve regeneration. PEO anodization did not alter filament degradation after six weeks in vivo, nor did it improve nerve regeneration after 14 weeks in vivo. However, polishing slowed Mg degradation in vivo compared to as-received filaments and improved growth of axons across the conduit gap. This study provides further evidence that Mg is well tolerated in vivo for supporting nerve regeneration and is one more study showing a difference in degradation rates between in vitro and in vivo exposure. Although PEO anodization did not slow degradation in vivo, our study shows support for the hypothesis that slowing the degradation rate of Mg could benefit nerve regeneration.

## 2. Materials and Methods

### 2.1. Mg Filaments: Source, Cleaning, and Polishing

As in our previous work [10,11], pure Mg (99.9% purity) filaments of 250 +/− 25 µm were purchased from Goodfellow USA Corporation (Coraopolis, PA). Impurities included <1 ppm Ag, 10 ppm Ca, 5 ppm Cu, 300 ppm Mg, <1 ppm Na, 100 ppm Si, and 100 ppm Fe. The iron content is above typical tolerance limits, thus leading to the expectation of some micro-galvanic action [21].

Methods for preparation and anodization were similar to ones used previously for treatment of an AZ31 Mg alloy [22,23]. To prepare as-received (Unpolished (UnP)) Mg filaments, segments were cut from the as-received coil, and straightened by hand. Segments of 2 cm were used as-is for nerve repair experiments. For anodization, segments were placed in 200 g/L chromium trioxide (CrO_3_) and 10 g/L silver nitrate (AgNO_3_) for 1 min, which cleans the surface without significant Mg corrosion, as has been reported [24,25]. Samples were then hand polished with 1200-grit abrasive paper, using isopropanol as a lubricant, to produce “Polished (Pol)” filaments. All filaments were ultrasonically cleaned in ethanol. All materials were from Thermo Fisher Sci., Waltham, MA, USA, if not otherwise specified.

### 2.2. PEO Anodization

Polished (Pol) Mg filaments to receive PEO coatings were placed in micro-needles (diameter 1 mm) as a holder and ~8 mm of Mg filament was exposed. Epoxy was used to create a watertight seal between the Mg filament and micro-needle for coating. Following the PEO coating, any excess coated Mg filament was cut so that the remaining length was 5 mm (±1 mm). Samples of the Pol and polished and anodized (PEO) filaments that were cut to this length had a surface area (SA) of 4.03 mm^2^ and volume of 0.245 mm^3^.

AC PEO treatment was conducted in an electrolyte solution after samples were carefully rinsed with ethanol and dried. The basal electrolyte solution consisted of 0.25 M NaOH + 0.04 M NaH_2_PO_4_•2H_2_O, which was prepared using deionized water and was continuously stirred by a magnetic stirrer during treatment. The coatings were formed in a 500 mL glass container with a stainless steel rod as a cathode. At room temperature, pulse waveforms were applied using the EAC-S 1-phase AC power source (EAC-S250F/F1000/USB/SD) manufactured by ET System^®^ Electronic GmbH (Altlussheim, Germany). The AC PEO treatment was conducted with the treatment parameters of 100 V RMS, 100 and 250 Hz for 5 min. Voltage/current responses were monitored electronically with a Tektronix TDS 2020 oscilloscope (Tektronix, Inc. Beaverton, OR USA). After coating, samples were rinsed with deionized water and dried in air. The anodization conditions were the same as used previously by our group to study the effects of different electrolytes on anodization of a Mg alloy [22,23].

### 2.3. In Vitro Immersion Testing of Mg Filaments

In vitro corrosion studies were performed using an immersion solution of Hank’s Balanced Salt Solution (HBSS) (HyCloneTM HBSS w/o Phenol Red Lot No: AXJ46328, purchased from Thermo Fisher Sci.). HBSS contains all the physiological inorganic salts at concentrations similar to those in the human body and, while not completely equivalent to extracellular fluid composition, is the most commonly used medium in the Mg in vitro literature and provides a base solution to allow comparisons between the reported results [26]. The pH value of HBSS as purchased was 7.4 ± 0.005 and no adjustments were necessary. The Mg filaments were placed inside 15 mL polypropylene tubes (Falcon^®^ Blue Max 352097)) and 5 mL of HBSS was pipetted into each tube. The sample medium volume to sample surface area ratio was chosen to be as large as practically possible, to mitigate secondary effects stemming from small solution volume [6,27]. The volume to surface area ratio was 5 mL/0.04 cm^2^ and samples were placed inside an incubator and maintained at 37 °C and 5% CO_2_ for 1–28 days. The pH values of the solution were recorded every 48 h and the HBSS solution was changed every week. The test tubes were closed during incubation and no signs of contamination were observed.

### 2.4. Structural Characterization of Filaments Using Scanning Electron Microscopy (SEM) and Energy-Dispersive X-ray (EDX)

The surface or cross-sections of the filaments were observed after materials were dried and sputter-coated with a thin gold/palladium layer (Bio-Rad^®^ E5400 sputter coater, Biorad/Polaron, Cambridge, MA) to improve conductivity. Specimens were imaged using a scanning electron microscope (SEM), operated at a voltage of 5 kV and probe current of 10 mA (SI8000, Hitachi, Ltd., Hitachi, Japan). Elemental composition of the surfaces was analyzed using energy-dispersive spectroscopy (EDX) (Bruker AXS5350, Berlin, Germany). For cross-sectional analysis, the Mg filaments were first mounted in a two-part epoxy (Epokwick^®^ epoxy resin, Buehler, IL, USA) and allowed to cure. After curing overnight, samples were mechanically cut in half using a rotating saw (Allied^®^ TechCut 5, Allied High Tech Products, Inc., Rancho Dominguez, CA, USA). The ends of the samples were not studied for cross-sectioning in order to eliminate edge effects that occur during the coating process. Cross-sections were prepared by hand polishing with 600–1200 grit SiC paper. A representative section of good quality (approximately halfway through the filament) was chosen from 4 time points (7 days, 14 days, 21 days, 28 days) for detailed analysis.

### 2.5. Characterization of Mg Degradation Using Micro Computed Tomography (micro-CT)

For micro-CT scans of Mg filament for analysis of in vitro degradation, samples were placed on a rotatable stage between the imaging system (detector) and the radiation source, using a Nanotom CT (Phoenix Nanotom-MTM, GE^®^ Sensing & Inspection Technologies GmbH, Wunstorf, Germany). X-ray scans were performed at 30 kV and 60 μA with 800 images taken with voxel size of 2.3 μm. The image reconstruction software VG Studio Max (v 2.1, Volume Graphics, Charlotte, NC USA) was used to reconstruct 3D models of Mg filament after the acquired data in PCA (Phoenix CT Acquisition) file format were converted to PCR (Phoenix CT Reconstruction) format.

### 2.6. Cytocompatibility of Mg Filaments Using Cell Cultures

Primary porcine tracheobronchial epithelial (PTBE) (airway cells) were provided by Dr. Jenora Waterman (North Carolina A&T State University, Greensboro, NC USA) [28]. Frozen aliquots of cells were placed in pre-warmed media (50:50 mixture of Dulbecco’s Modified Eagle Medium (DMEM) and Ham’s F-12 containing 2% fetal bovine serum (FBS), antibiotics and growth supplements) and pipetted gently to evenly disperse cells. Cells were seeded at a density of 3.05 × 10^5^ PTBE cells/well (in a volume of 1 mL media/well, 12 well dishes). Filaments were categorized into three test groups: titanium filament (Ti, not shown), Pol Mg filament, and PEO-anodized Mg filament. Filament samples were mounted on glass cover disks using small amounts of polydimethylsiloxane (PDMS, Sylgard^®^). Three samples were mounted per disk and 2 disks were used for each group. Samples were sterilized (UV light, 15 min); cells were added and incubated under standard cell culture conditions (100% humidity, 37 °C, 5% CO_2_) for 12 h. Following incubation, the medium was aspirated, and samples were rinsed three times with 1X phosphate-buffered saline (PBS) pH 7.4 to remove non-adherent cells.

To visualize live and dead cells attached to metals, Live/Dead staining was performed using a dye solution (EthD-1 (4 μM) and Calcein AM (2 μM) in PBS). PBS was removed from the cells, the staining solution was added, and wells were kept in the dark for 30 min. Then, samples were flipped over inside the wells and imaged with an EVOS^®^ FL cell imaging system (Thermo Fisher Sci.).

After imaging, samples were rinsed in PBS (to remove non-attached cells), fixed in 100% methanol, dehydrated successively (5 min washes of 70%, 2 × 95%, 100% ethanol, and one 30 min wash in 100% ethanol), dried, and imaged by SEM as above.

### 2.7. In Vivo Nerve Repair

#### 2.7.1. Animals and Surgery

A total of 40 adult female Lewis rats were used and all protocols were approved by the University of Cincinnati (UC) Institutional Animal Care and Use Committee (IACUC), according to the NIH Guide for Use of Animals. Procedures were as previously described [11]. In brief, animals were anesthetized with isoflurane and administered analgesics. The upper thigh area of one leg was prepared for sterile surgery. The sciatic nerve was exposed, a section of nerve was removed, the ends were allowed to retract, and repairs were made resulting in a gap ~15 (15–16) mm long. For the positive control, gaps were repaired with isografts (Iso, positive control), where nerves from a donor rat of the same species, sex, and age were reversed and sutured (10–0 sutures) to nerve stumps. For experimental groups, repairs were with hollow nerve conduits with or without (negative control) a single metal filament. Nerve conduits were made of poly (ε-caprolactone) (PCL), prepared using a dipping method as previously described (generously provided by D. Minteer and K. Marra) [10,11,29]. Conduits were 18 mm long with ~0.23 mm wall thickness and ~1.5 mm internal diameter. Metal filaments (Mg or Ti) were 250 µm diameter and 2 cm long and sterilized by sonication in ethanol followed by 40 min of exposure to UV light. Filaments were placed within a conduit and both ends were embedded ~2 mm into each nerve stump. Nerve stumps were pulled into both ends of conduits by ~1 mm and sutured to the nerve epineurium with 10–0 sutures. All conduits were filled via syringe with sterile 0.9% saline. Muscles and skin were closed with 4–0 sutures and animals were warmed until mobile, then returned to their cages.

The experimental groups were repairs with (1) isografts (Iso) or conduits containing (2) no filaments (Empty/Em), (3) Mg filaments as received (Unpolished/UnP), (4) Mg filaments acid treated and polished (Polished/Pol), (5) Mg filaments PEO anodized (PEO), and (6) titanium filaments (Ti). Each group had *n* = 6 animals, except for the UnP group where, due to technical reasons, there were *n* = 4 animals. No animals showed ill health or autophagy at any time until euthanized after 14 weeks via UC IACUC-approved protocols.

#### 2.7.2. Micro-CT Imaging

Micro-CT imaging was performed using live animals at 6 weeks post-surgery (6 WPS) and of tissues after euthanasia at 14 WPS. For 6 WPS live animal imaging, using IACUC-approved protocols, animals were anesthetized (isoflurane) and body temperature was maintained while their injured legs and pelvis were imaged. They were returned to their home cages afterwards. For 14 WPS imaging after euthanasia, isografts or conduits with ~2 mm of both nerve stumps were removed, immobilized to a solid support, and fixed (4% paraformaldehyde in PBS, 24 h). Fixed nerve tissues were rinsed (5×) in PBS, incubated in a 2% Lugol’s iodine solution (diluted Lugol’s, 6% total iodine, Thermo Fisher, room temperature) for 48 h, rinsed in PBS (until solution was almost clear), and imaged by micro-CT, as previously described by our group [30,31]. Imaging was performed using a Siemens Inveon Multimodality System (San Diego, CA, USA) at a UC Core Laboratory. Stationary samples were scanned at half-degree increments with 384 steps (step and shoot) for 192 degrees. Images were acquired with high magnification and a pixel matrix binning of two, resulting in an effective voxel size of 17.27 microns, using 80 kVp voltage and 300 µA current, with the exposure time at 2100 ms with 25 ms settle time.

With the micro-CT image sequences at 6 WPS, quantitative analysis of the SA and V of remaining metal was performed using a Siemens Inveon software package. Filament pieces were included if they had radiodensities in Hounsfield units (HU) between 620 and 4070 HU and were separated from bone. The SA and V values were strongly and positively correlated at 6 WPS (r^2^ = 0.95 with *p* < 0.0001, Appendix A). SA to V (S/V) ratios at 6 WPS were also calculated (Appendix A). Prior to surgery, all Pol and PEO filaments and one UnP filament were imaged by micro-CT. The average SA was 20.6 +/− W 0.5 mm^2^ and V was 1.35 +/− 0.06 mm^3^. In the UnP group, 3 animals received unimaged filaments due to loss of filaments during surgery, and to calculate degradation rates, these were assigned the average V for all other filaments (above). Degradation rates (DRs, in mm/year) at time I (6 WPS) were calculated using the following formula, based on a previous study of Mg pins implanted in vivo [32]:(1)$$DRi=\frac{\Delta \mathrm{xi}}{\Delta t}\;with\;\Delta \mathrm{xi}=\frac{\Delta \mathrm{Vi}}{\mathrm{Si}}$$

Inveon software was also used to render images in 3D and, for 14 WPS samples, to export image sequences as DICOM format files, which were viewed and analyzed with the public domain image JAVA processing program, ImageJ.

#### 2.7.3. Animal Behavioral Testing for Nerve Regeneration

At regular intervals, animals were weighed, and non-invasive behavioral tests were administered, as described previously [10,11], with some tests adapted from the literature [33]. In brief, (A) muscle atrophy was monitored in live animals by measuring the external circumference of both hind legs at the mid-belly level of the calf muscle (see Appendix A). (B) Return of sensory function was analyzed using a pinch test of the injured hind foot and toes (Appendix A). Reactions of toes #2–4 and reactions to pinches of the skin at the side of the foot were highly variable, likely due to gradual expansion of the medially located saphenous nerve territory [33,34]. (C) Return of motor function was measured by scoring reflex extension of lateral toe #5 (toe spread, scoring details in Appendix A). Values are injured vs. control leg as a percent (injured/control × 100).

#### 2.7.4. Electrophysiology

At 14–15 WPS, electrophysiological measurements were made of nerve connectivity across the injury gap. Using IACUC-approved protocols, animals were anesthetized (isoflurane) and the sciatic nerves on both legs were surgically exposed. Nerves were stimulated proximal to conduits on the injured leg and a similar position on the uninjured leg and the response of the triceps surae muscle was measured via an electrode positioned in the depth of the muscle. Electrical stimulation was increased until the muscle response plateaued, and measurements were made of the peak amplitude and peak area under the curve of the average CMAP (compound muscle action potential) per stimulation intensity and averaged. Velocity of nerve activity was measured. For each measure, reported values are for injured normalized to control leg per rat as a percent. After recording, animals were euthanized without recovering from anesthesia, using IACUC-approved protocols.

#### 2.7.5. Sacrifice, Tissue Processing, Iodine Infiltration

After euthanasia, isograft nerves, control nerves, and conduits plus ~2 mm of proximal and distal nerve stumps were dissected, attached to a physical support to keep tissues straight, and fixed in ice-cold 4% paraformaldehyde, pH 7.4 for 24 h. After nerve removal, the triceps surae (both heads of the gastrocnemius muscle and the soleus muscle) of both hind legs were carefully dissected out and weighed immediately. After fixation, nerve tissues were treated with iodine and imaged by micro-CT as described in Section 2.7.2. After imaging, tissue iodine was removed by incubation in 2.5% sodium thiosulfate in PBS for 48 h, as described previously [30,31], rinsed (PBS), and embedded in paraffin using standard procedures.

#### 2.7.6. Histology

Paraffin-embedded tissues were sectioned (7–10 µm thick) in sets taken at equally spaced intervals along the entire length of the tissues. Slides from each set were stained with hematoxylin and eosin (H&E, coverslipped with Permount) and adjacent slides were immunostained using procedures described previously [11]. Details included blocking buffer (1 h), overnight primary antibody incubation, and two-hour incubation in secondary antibody at room temperature with three PBS rinses between steps. Axons were labeled with rabbit anti-neurofilament protein 200MW (NF) (1:500 dilution, Sigma-Aldrich, St. Louis, MO, USA) and the secondary antibody was goat anti-rabbit Alexa 594 (1:100 dilution, Thermo Fisher Sci.). Then, all slides were stained with 4′6-diamido-2-phenylindole (DAPI, 1:1000 dilution, Sigma-Aldrich, St. Louis, MO, USA) and coverslipped with Fluoromount. Analyses of histology had one less animal in the Pol group (*n* = 5) due to tissue damage during processing and the Ti group was not sectioned due to hardness of the undegraded metal.

#### 2.7.7. Microscopy and Photography

Photographs of immunostained tissue sections were taken on a wide-field upright Zeiss Axio-plan Imaging 2e fluorescence microscope (Zeiss, Jena, Germany) using grayscale images captured by a QICam cooled CCD camera (Q Imaging, Burnaby, British Columbia, Canada) and channels of immunostaining were combined and pseudo-colored using Photoshop. Autofluorescence in the green channel was also imaged. Color images of H&E staining were taken using a Zeiss Axiocam digital camera (Zeiss). Analysis of relative amounts of NF staining in nerves was performed with ImageJ using subjective scale (see Appendix A). Photoshop was used to prepare figures and brightness and contrast were enhanced to illustrate features, but this did not impact quantitation.

### 2.8. Statistics

All analyses were performed by personnel blinded to conditions and used random sampling methods where applicable. Analysis of functional behavior was carried out by one person to maintain consistency, while analysis of images was realized by two or more personnel, independently. For the Calf Circumference data, a linear mixed-effects model was used with the SAS software to assign significance, after which multiple comparisons between differences of least squares means were analyzed after applying the Tukey–Kramer adjustment, using *p* < 0.05 for significance. For dissected muscle weights, an ANOVA analysis was performed using SAS with the Dwass, Steel, Critchlow–Fligner multiple comparisons test. For all other data analysis, SigmaPlot software (Sysstat Software Inc., San Jose, CA, USA) was used to test normality, and if normal, data were analyzed with Student’s *t*-test or ANOVA, with the Holm–Sidak multiple comparison test. For data not distributed normally, a Kruskal–Wallis ANOVA was used, followed by Dunn’s post hoc test. Significance was assessed at *p* < 0.05. Column graphs show mean +/−SD; box graphs show median and 25/75 percentiles.

## 3. Results

### 3.1. Mg Filament Preparation and Anodization

Mg filaments, as received from the manufacturer (UnP group), had surface deposits as shown by SEM (Figure 1a). Characterization by EDX (Figure 1b–d and Table 1) showed that the surface layer was higher in atomic % of oxygen than Mg.

As surface deposits could interfere with anodization and cellular interactions, and could alter degradation rates, the Mg filaments were cleaned by acid treatment and polished (Pol group). These treatments removed significant amounts of the deposits (Figure 1e), and EDX characterization showed removal of most of the oxides, with more Mg available at the surface (Figure 1f–h, Table 1).

### 3.2. Anodization and In Vitro Degradation

Surface smoothness of Mg filaments was characterized by SEM analysis. Figure 2a,b shows that polishing removed significant amounts of surface degradation products, resulting in a smoother surface. Anodization by PEO resulted in a similar smooth appearance (Figure 2c).

The Pol and PEO filaments were immersed in a physiological salt solution (HBSS, 37 °C) for 4 weeks. The filaments were imaged weekly by micro-CT (Figure 3a,b), light microscopy (Appendix A), and SEM after 4 weeks (Figure 3c–f). By micro-CT (Figure 3a,b), the Pol filament showed a progressive loss in length beginning at week one, while the PEO filament did not lose length until 4 weeks. By light microscopy (Appendix A), corrosion of the surface was visibly less for PEO than Pol at week 1, with corrosion on both after that, but with greater loss of structure with Pol. By SEM imaging after 4 weeks (Figure 3c–f), the Pol filament (vs. the PEO filament) had greater surface corrosion, including pitting and cracking. Micro-cracks were seen on the surface of each type of filament, filled with corrosion products, but were more uniform in size and shape for the PEO filament. It should be noted that some of the cracking can be attributed to drying after removal from immersion fluid and preparation for SEM. 

To further examine the corrosion layers, cross-sections of Pol and PEO filaments, before and at each week after immersion, were examined by SEM with EDX analysis. Selected images are shown in Figure 4a–f, while the complete set of images are shown in Appendix A (Pol) and Appendix A (PEO). Before immersion, SEM imaging (Figure 4a,b) showed that anodization deposited a layer that appears as a thin bright line in SEM and by EDX contained oxygen and phosphorous (oxides and phosphates) (Appendix A). This layer was determined to be 2 µm thick. After 4 weeks of immersion in HBSS, the Pol filaments (Figure 4c,e and Appendix A) had a smaller area of pure Mg and a thicker deposition layer over most of the filament in comparison to PEO filaments (Figure 4d,f, and Appendix A). The corrosion layers for both were composed of Mg complexed with oxygen, calcium, and phosphorous. EDX point analysis was conducted at 5 points on the surface and values were averaged to provide the data shown in Appendix A. The Ca/P ratio of 1.53 ± 0.18 for the uncoated sample suggests calcium-deficient hydroxyapatite (CDHA), which is 1.5–1.67, while the Ca/P ratio of 1.76 ± 0.17 for PEO coated sample suggests hydroxyapatite (HA), which has a ratio of 1.67 [36,37]. While variable depths of corrosion and pitting were seen in both filament types, the PEO filament had a more uniform and thinner corrosion layer overall, and regions of pitting corrosion were not as deep as in the Pol filament. Notably, EDX analysis showed that for both, the corrosion product layer had an outer shell of calcium phosphate (CaP) and an inner layer of oxides, primarily Mg(OH)_2_, which would be denser than CaP and presumably prevented inner diffusion of calcium and phosphates. Over time, it was not possible to distinguish between the PEO coating and corrosion products as the coating contained phosphate.

The thickness of the Mg(OH)_2_ and CaP layers, separately, was measured at 10 different sites per image. Figure 4g shows that the depth of the Mg(OH)_2_ layer was greater for the Pol filament than the PEO filament from week 1 on. For the CaP layer (Figure 4h), the Pol filament showed thicker measurements at 2 and 3 weeks but did not continue to increase at 4 weeks. This might be explained by the peeling/flaking away of some of the CaP layer on the Pol filament as it became thicker. The CaP material did not adhere well to the inner layer, as seen in the insert in Figure 4c (white bar illustrates one measurement). If immersion had been continued past 4 weeks, the PEO sample might also have shown CaP layer flaking. Overall, the data show that the PEO coating provided significant protection from corrosion over the 4 weeks of immersion, but this protection started eroding after week 3.

### 3.3. Cytocompatibility: Cell Attachment to Polished and Anodized Mg Filament

Cell attachment and cytocompatibility were determined using an epithelial cell type that is one of the most prolific in the body, primary porcine tracheobronchial epithelial cells (PTBE). Cells were plated onto pieces of filaments adhered to the substrate with epoxy. Figure 5a,c shows SEM images of Pol and PEO Mg filaments immersed in growth medium for 12 h without cells, indicating some increase in the surface roughness of the PEO substrate. With addition of cells (Figure 5b,d), both surfaces supported roughly similar cell attachment, with cells well spread out, multiple points of attachment to the surface, and well-extended filipodia attached to the surfaces. With a Live/Dead cell assay (live cells fluoresce green, dead cells red), fluorescent light microscopic images (Figure 5e–h) showed that greater numbers of living cells attached to the PEO compared to Pol filaments (not quantified). Control Ti filaments showed no attachment of any cells (not shown). Thus, the PEO anodization improved cell attachment and viability over Mg filaments alone.

### 3.4. In Vivo Use of Mg Filaments for Peripheral Nerve Repair

#### 3.4.1. Mg Filament Degradation In Vivo

At 6 WPS, live animals with Mg filaments were imaged by micro-CT to determine degradation status of filaments. Representative filament images rendered in 3D are shown in Figure 6a–c, and SA quantification in Figure 6d. There were gaps in all filaments.

Significant differences were detected in SA between groups (Figure 6d, ANOVA, *p* = 0.038, *n* = UnP (4), Pol (6), PEO (6)). The Pol group was significantly higher than UnP (Holm–Sidak, *p* = 0.041), but the PEO group was not different from either other group (Pol vs. PEO: *p* = 0.162; PEO vs. UnP: *p* = 0.262). The PEO group had the greatest variability in SA (and V, SA to V correlation in Appendix A). Of six PEO animals, two had filaments with the smallest SAs, three had mid-range values, and one filament had the largest, circled in red in Figure 6d (see also Appendix A to interpret symbol overlap in 6d).

The filament with the greatest SA in the PEO group was a unique case, as determined by comparing 6 WPS micro-CT images (Figure 6e) with 14 WPS micro-CT images of tissues after euthanasia, with soft tissue contrast (Figure 6f). At 6 WPS, the filament was unique among all in that the Mg pieces were semi-adjacent (red arrows inside red circle in Figure 6e), suggesting filament breakage. At 14 WPS, images from the same animal showed denser materials that appeared to be Mg degradation products in a pattern that matched the six-week images (Figure 6f, red arrows inside red circle). The conduit was also severely broken (yellow lines mark some conduit pieces) around these pieces and had a complete proximal break (green vertical arrow in 6f). The images suggest significant interference with tissue regeneration, which was confirmed later in histological sections, where there were (uniquely) no axons distal to the repair. Without this animal’s value, the ANOVA on SA was still significant (ANOVA on ranks, *p* = 0.023), with the same post hoc conclusions (Pol significantly higher than UnP (*p* = 0.042) with the PEO group not different from other groups (PEO vs. Pol: *p* = 0.094, PEO vs. UnP: *p* = 1)). Overall, our data showed that polishing slowed the degradation rate in vivo, but PEO anodization did not slow degradation further.

Degradation rates (DRs) and SA to V (S/V) measurements were calculated as in a previous study [32]. The DRs for the 6 WPS time point showed significant variability for the UnP and PEO groups due to the variability in V, with very high DRs (Appendix A) of around 6 mm/year for two UnP animals and around 11 mm/year for two PEO animals (symbols overlap in Appendix A). The average DR for all other animals was 1.14 +/− mm/year (range 0.64 (the animal in Figure 6e) to 2.0 mm/year). Given this uneven distribution, an ANOVA on ranks showed significant differences in DRs between groups (*p* = 0.05) and Dunn’s post hoc test vs. UnP showed Pol (*p* = 0.037), but not PEO (*p* = 0.742) different from UnP. Calculating S/V gives an idea of how rough the surface of the filaments is after degradation. As seen in Appendix A, S/V was not different between the groups (ANOVA, *p* = 0.475). DRs were loosely related to S/V values (Appendix A).

#### 3.4.2. Function

Animals were weighed and functional scores were assessed at regular intervals, as detailed in Figure 7. Weights were not different between groups (not shown). Motor function was assessed by analyzing reflex lateral extension of the lateral toe (toe spread) on the injured foot (Figure 7a, subjective scale in Appendix A). Due to the ordinal scale and unequal variance, statistical analysis was not possible, but the data show that only the Iso group showed consistent changes beginning at WPS 6, and becoming consistently higher, with scores of 2 or 3 for all Iso animals between WPS 10 and 13 (but never scores of 4 = normal). For all conduit groups (Ti, Em, UnP, Pol, and PEO), scores were 0 over all time points, with a few rare exceptions that were not considered above subjective variation.

Sensory function was measured by a pinch test (Figure 7b, scale in Appendix A and Appendix A, no statistical analysis). By 1 WPS, all animals had lost sensation except at the ankle (score of 1). Between 3 and 6 WPS, sensation returned for skin on the lateral side of the foot (scores 2 and 3) in all groups. Small group differences during these weeks were highly variable. By 7 WPS, all animals had regained lateral foot sensation up to the base of the lateral toe (score of 3), attributed to sprouting of medial foot innervation, not sciatic nerve regeneration (see Methods). After 7 WPS, sensation to the lateral toe (a score of 4, considered to be more indicative of regeneration) returned for many Iso animals, and by 12–14 WPS, all Iso animals had scores of 4. For all other animals, scores of 4 were rare between 12 and 14 WPS.

Muscle atrophy occurs when motor nerves are cut and can be reversed with regeneration. In live animals, this was assessed by measuring leg circumference at the belly of the calf muscle (see Appendix A for measurement method and Figure 7c, injured/control leg circumference as a percent). Control leg circumference did not differ between groups at any time point (ANOVA/data, not shown). Statistical analysis showed that the Iso group had a slower loss of calf circumference, significant vs. most groups between WPS 1 and 4 (see Figure 6 legend for specifics). All groups had similar measurements at WPS 3 and 6, with 6 being a common plateau, with the injured leg at ~60% size of the uninjured leg. After WPS 6, only the Iso group showed increased calf sizes that were significantly greater than all other groups between WPS 10 and 14. There were no differences between the conduit groups at any time point. At WPS 6, all animals reached a common low value of around 60%. By comparing values to the common plateau at WPS 6, the Iso group increased calf size at WPS 10–14. In the conduit groups, UnP, Pol, and Ti groups did not change compared to WPS 6, while the Em and PEO groups decreased in calf size at WPS 13 for Em and WPS 13 and 14 for PEO. Thus, the Iso group showed significant recovery compared to WPS 6, while the conduit groups plateaued the WPS 6 values or continued to atrophy.

Muscle atrophy was also quantified by measuring wet weights of carefully dissected calf muscles after euthanasia (Figure 7d, weight of injured/control muscle as a percent). The Iso group percentages were significantly higher than all groups except the UnP group (ANOVA on ranks, *p* < 0.001, SAS software).

#### 3.4.3. Electrophysiology

Nerve integrity across the injury site was assessed by stimulating the nerve above the site and measuring electrophysiological action in the calf muscle. As shown in Figure 7e–g, the Iso group was significantly higher than all others in CMAP peak amplitude (ANOVA, *p* < 0.001, Holm–Sidak post hoc test) and CMAP peak area (ANVOA on ranks, *p* = 0.015, Dunn’s post hoc test comparing all to the Iso group). Conduction velocity was not different between the groups.

#### 3.4.4. Micro-CT Imaging with Iodine Contrast of Nerve Tissues

Nerve tissues removed after euthanasia at 14 WPS and imaged with micro-CT with iodine for soft tissue contrast were analyzed for conduit breakage and tissue regeneration. Images from one PEO animal are shown in Figure 8a–e. The CT imaging showed multiple breaks in all conduits, to varying degrees, in both longitudinal 3D reconstructions (constructed with ImageJ, Figure 8a) and in axial single images (Figure 8b–e), and the images suggested both lateral and longitudinal compressive forces. Tissues inside the conduit are outlined between the white arrows in Figure 8a (longitudinal view). Green arrows in Figure 8a,c (one axial view) point to an area considered to be one of the thinnest regions of the tissue strand. Analysis was performed of the soft tissues at points judged to be narrowest to give a subjective tissue continuity (TC) score (described in Appendix A). Figure 8m shows the TC values between groups. Omitting the only animal with a TC score of 0 (the animal in Figure 6e–f), there were significant differences between groups (Figure 8m) (ANOVA, *p* = 0.035), but no groups were significantly different by the Holm–Sidak post hoc test, although the Ti group was higher than PEO at *p* = 0.059.

#### 3.4.5. Histology of Nerve Tissues Compared to Micro-CT

After CT imaging, removal of iodine by incubation in sodium thiosulfate, and paraffin embedding, cross-sections of nerve tissues were cut at multiple points. Alternate sections were stained with H&E or immunostained for axons (anti-neurofilament 200 MW protein, NF) and nuclei (DAPI). For the same Pol animal shown in Figure 8a–e, stained histological sections were matched to selected micro-CT cross-sections and confirmed extensive breakage of conduits. Sections stained for H&E (Figure 8f–h) and immunostained for axons (red, anti-NF, Figure 8i–k) confirmed the presence of myelinated axons at the distal ends of the conduits for all animals, except the one shown in Figure 6e–f. The NF staining was subjectively scored (scale detailed in Appendix A). The NF score (Figure 8l) was not significantly different between the Em, UnP, Pol, and PEO groups (normally distributed, ANOVA, *p* = 0.556).

#### 3.4.6. Comparisons between Filament Degradation at 6 WPS and NF Scores at 14 WPS

As we sought in this study to determine if the rate of Mg degradation had an effect on nerve regeneration, we compared the NF scores at 14 WPS of the three Mg groups with the SA and DR values determined by imaging filaments at 6 WPS. As shown in Figure 8n, the 14 WPS NF scores were significantly and positively correlated with the 6 WPS SA (r^2^ = 0.38, *p* = 0.019). The NF scores were negatively correlated with the 6 WPS DRs (r^2^ = 0.497, *p* = 0.0048) (Figure 8o). This shows a relationship between the extent of filament degradation by 6 WPS in vivo and the amount of axonal regeneration across the injury gap as analyzed at 14 WPS.

## 4. Discussion

Mg metal is an exciting biomaterial for biomedical implants because of its in vivo biosafety and safe degradation (bioabsorption). However, its degradation rate can be highly variable in vivo due to different degrees of water exposure between implant sites. This makes it challenging to adjust the degradation rate to the clinical need for the implant purpose. In addition, the fact that rapid degradation of Mg releases H_2_ gas and can locally raise pH must be considered.

We have uniquely been exploring the use of Mg metal filaments to provide physical support and contact guidance to peripheral nerve axons as they regenerate across an injury gap in nerves. Mg metal might also release Mg ions during degradation that are beneficial to the recovery of nervous tissues after injury. In peripheral nerve repair, clinical challenges arise due to the need for surgery and tissue replacement after segments of nerve are lost. To replace the use of autografts, biomaterial hollow nerve conduits have been well studied and several are in clinical use [3,5]. However, these are successful only for gaps less than a critical length of 20–25 mm in humans, or 14–15 mm in adult rats [6]. Preclinical research indicates that nerve regeneration across critical gaps can be improved by many factors, one of which is the addition of linearly oriented materials inside hollow nerve conduits [5,38,39,40]. Mg metal has promise for this internal linear support because of its degradability and safety. Safety has been shown by the fact that Mg metal implants have been used in human patients in Europe and Asia for >10 years. In 2021, the safety of a Mg metal cardiovascular stent was shown in >1000 patients after one year of implantation, while in 2022, safety and efficacy were shown for another Mg stent in 207 patients after one or two years of implantation [41,42]. In terms of the Mg ions released during metal degradation, Mg salt solution administration in preclinical studies is known to improve the recovery of damaged brain tissues after strokes or traumatic damage [43].

Our previous studies explored peripheral nerve repair using a single Mg filament placed inside a PCL conduit and we saw improved axonal content distal to non-critical gaps (6 mm), but not 15 mm gaps [10,11]. One possibility for the lack of effect with longer gaps could be that the Mg metal underwent earlier degradation due to longer exposure to corrosive salt solutions inside conduits, before being covered by cells. Therefore, we sought a method to slow the Mg degradation rate.

### 4.1. Effects of Polishing on Mg Metal

Mg filaments received from a commercial source had a layer of corrosion products that consisted primarily of oxides and resulted in little Mg exposed to interact with cells or tissues. This surface was rough and would have interfered with anodization, so it was removed with acid treatment and polishing. The resulting surface was smoother and had a higher Mg content exposed. Removal of this surface material reduced the in vivo degradation rate significantly, as is further discussed below.

### 4.2. PEO Anodization

To reduce the corrosion rate of Mg, researchers have generated different Mg alloys or coatings. Anodization, as discussed previously, results in a highly adherent ceramic oxide coating [44] with sufficient wear and corrosion resistance, desirable electrical properties, and high thermal stability [17,18,19,45]. MAO coatings have been shown to slow degradation rates and reduce hydrogen evolution upon implantation in vivo [44].

Factors that must be considered with anodization are the electrolyte solution used to provide distinctive features to the coatings. The electrolyte elements exert a decisive influence on the character and features of the ceramic coating, while the electrolyte chemistry influences the nature of the pores formed in the ceramic coating and pore size distribution as well as the phases present in the coating [19]. The salt solution chosen for anodization was designed to ensure biocompatibility with nerve regeneration. Phosphorous has been studied for PEO coatings, where it is readily converted into a calcium phosphate complex, which has been shown to improve cell attachment as well as improve the corrosion resistance of Mg [14,32]. Thus, an electrolyte solution containing phosphorous was used to PEO coat pure Mg. In our previous studies, we tested solutions containing Si and F with AZ31 Mg [22,23]. The incorporation of KF yielded (vs. other electrolytes) thicker coatings, a microstructure with fewer pores, a smoother appearance, a higher Weibull modulus (lower degree of scatter) for both hardness and Young’s modulus compared to other coatings, and the highest corrosion resistance. All PEO coatings protected samples from corrosion during immersion in HBSS compared to uncoated samples.

For nerve repair, we used pure Mg, because the AZ31 Mg alloy contains aluminum, which has been shown to be neurotoxic both in vitro and in vivo at very low concentrations or levels of ingestion [46,47,48]. Similarly, neurotoxicity was seen with nanoparticles of Al, Cu, and Ag [49]. We also avoided Si and F as electrolytes because, even though studies have shown promising effects with cultured cells of PEO coatings including Si and F, no studies have explored neurotoxic effects, especially with delicate regenerating nerve tissues [14]. Even if Si and F did not have ill effects in vivo, they could be bioactive and have complicating factors. Therefore, we chose pure Mg and an electrolyte solution containing only sodium, oxygen, hydrogen, and phosphorus. This highlights a critical aspect of anodization; it needs to be employed with the application site in mind and considering the possible adverse effects of the ionic species that will bloom at the coating surface.

PEO is an anodization method performed at voltages higher than the dielectric breakdown of the oxide layer that is formed on the metal surface. The high temperatures (30,000 °F) and sparking phenomena that develop cause the surface metal to partially melt and react with the oxygen that is present to create a porous, irregular surface that is a well-adhered ceramic oxide coating that forms a barrier layer due to the reaction between Mg^2+^ and anions in the solution [14,32,50]. The final thickness of the coating is dependent on the relative rates of anodic dissolution and film formation, which occur simultaneously. For a thicker coating, the rate of film formation needs to be higher than the rate of dissolution. Our results after PEO anodization of pure Mg filaments resulted in a smooth coating of 2 µm, leading to reduced degradation via multiple analyses for up to 3 weeks upon immersion in a physiological salt solution compared to Pol filaments.

### 4.3. Effects of PEO Coating on Cultured Cell Attachment and Viability

Cells plated on Pol and PEO Mg filaments showed that the PEO coating improved cell attachment and viability over a period of 12 h. The PEO coating did not substantially alter cell spreading as cells were flattened and had multiple filipodia on both PEO and Pol filaments. This contrasts with reports that the roughness of the PEO coating causes cells to round up compared to uncoated surfaces [14]. This could have been because our coating was especially thin and smooth, as shown by the SEM images, with small pores and few cracks. This is consistent with our previous studies where the anodization layer on AZ31 Mg using a solution with Si and phosphate but not F gave the thinnest and smoothest surface [23]. We did not test cell attachment to UnP filaments, but as that surface was the roughest, it might have altered cell spreading towards a more rounded appearance.

### 4.4. In Vivo Mg Filament Degradation

After nerve repair, we analyzed Mg degradation by micro-CT imaging of live rats at 6 WPS. We chose 6 weeks to compare with our previous studies that showed substantial gaps in similar Mg filaments at 6 weeks [10,11]. By this time, we expected axons to have regenerated across the gap, based on both previous work and our own findings [10,11,51]. We showed that neither polishing nor PEO anodization slowed degradation sufficiently to preserve the full filament length; all three types of filaments had substantially degraded. Interestingly, polishing alone significantly slowed degradation compared to UnP filaments. One explanation might be that the roughness of the UnP surface with uneven deposition of degradation products vs. Pol led to greater pitting and galvanic degradation.

The PEO treatment did not slow degradation in vivo but, instead, resulted in high variability in degradation. Two PEO filaments had the smallest remaining sections at 6 WPS, which translated to the highest DRs of over 10 mm/year. In a study that examined MAO-anodized Mg alloy pins inserted (with uncoated controls) into the femurs of adult rats, a similar live micro-CT imaging process was used to follow pin degradation and also hydrogen gas creation over 12 weeks [32]. Their MAO coating slowed degradation in vivo by around 3 weeks, but after that point, MAO-coated pins degraded faster than non-anodized pins. The initial effects of slowing degradation in vivo reduced hydrogen gas production and improved bone deposition, which then resulted in adequate long-term bone repair. The authors suggested that the enhanced degradation of the MAO-anodized pins after 3 weeks in vivo was due to enhanced localized pitting corrosion at breaks in the coating and cut ends of the pins, which led to an irregular, rough surface (higher S/V ratios) that further increased pitting and galvanic corrosion. As in our study (Appendix A), they saw a suggestive, but not robust positive relationship between surface roughness (S/V values) and DRs (Appendix A). Degradation breaks were proposed at the depths of the pores and cavities formed within the highly convoluted surface created by anodization. In the study of MAO-coated bone pins, the total layer thickness was 10 µm but was around 2 µm (the thickness of our PEO coating) at the bottom of pores; these areas plus the cut ends of pins (which we also had in our filaments) were speculated to be sites for rapid pitting degradation [32].

Confounding factors in our analyses are, first, low animal numbers. Then, the micro-CT imaging at 6 WPS omitted small portions of the filament ends in a few animals because of difficulties in animal positioning. However, the data were examined for this effect and it was concluded that the errors would not have altered the findings. Other technical factors throughout further lowered numbers for some assays. However, the analyses shown had the appropriate statistical power.

### 4.5. Functional Effects of Metal-Containing Scaffolds on Nerve Repair

Overall, the functional tests of nerve repair showed that only the Iso group produced significant functional nerve regeneration, while none of the conduit groups did. This included function measured by toe spread, responses to a pinch, slowing of muscle atrophy after nerve transection, reversal of atrophy, and electrophysiological recordings of nerve integrity through the injury area. Between the conduit groups, there were very few differences. Some variability was seen in how soon calf circumference decreased, which may have indicated some physiological processes. All conduit groups reached a lower level and plateaued sooner than the Iso group, at 4 WPS vs. 6 WPS, which suggests conduit repairs in general were more damaging. Then, during the initial atrophy phase, the Pol group was similar to the Iso group at several time points, which is interesting. However, the differences were not consistent nor maintained over time as in the Iso group. None of the conduit groups showed an increase after week 6, as in the Iso group. Of interest was that the Em and PEO groups only exhibited a decline in leg circumference vs. the 6 WPS values, but these differences did not lead to intergroup variation at each time point, and no differences were identified in any other functional measure. In terms of muscle weights after euthanasia, the Iso group differed from most conduit groups statistically, but descriptively, it can be seen that all Iso group muscle weights were >30% (injured vs. control muscles) while all conduit groups had <30% values. The electrophysiology showed that only the Iso animals had nerve integrity through the gap that had innervated the calf muscles. A factor in the electrophysiological experiments was that some of the injured muscles were so thin that it was hard to get an electrode in, perhaps introducing some variability. Overall, the functional data show no consistent evidence that Ti or Mg filaments improved functional recovery compared to the Em group and only the Iso group evidenced improved function.

Nerve regeneration and functional recovery might have been affected by breakage of the conduits in vivo. Factors that could have been involved in conduit breakage were that the PCL conduits were relatively thin compared to our previous experiments (estimated at ~150 µm thickness vs. ~300 to 400 µm thickness previously), and this type of PCL conduit is produced by a dipping procedure that creates a relatively brittle material [11,29]. PCL also begins to degrade after implantation. The question then becomes, did the breakage occur before tissues grew across the gap or after? Our observations of NF expression distal to the conduits and continuous strands of tissue inside the conduits suggest that tissues and some axons were able to cross the gap before (or during) the breakage. However, this does not address the question of whether breakage interfered with function. Future experiments will need to test different types of conduits to rule out this factor.

The animal described in Figure 6e–f had an unusual amount of wire breakage, conduit breakage, and tissue disruption compared to other animals. In this animal, there was actual breakage of the filament at 6 WPS, and the micro-CT findings of tissues at 14 weeks confirmed that this was accompanied by conduit breakage. Blockage of regeneration by the breakage was confirmed as this animal had no NF staining or tissue continuity pathway through the conduit. The pattern of breakage is consistent with significant longitudinal compression of both conduit and filament between the knee and hip joints. This filament was imaged prior to implantation and it had the largest SA and V of any other filament (but was not significantly different by ANOVA) and it was placed in an animal with one of the lowest weights (data not shown). Thus, this might have resulted in greater compressive forces in this animal. We further speculate that the pre-6 WPS breakage compromised vascularization and therefore fluid exchange within the conduits, which then reduced clearance of Mg degradation products. Due to these factors, data from this animal were omitted from some analyses.

### 4.6. Analysis of the Effects of Differences in Mg Degradation on Nerve Regeneration

Despite the fact that PEO treatment did not slow degradation, the difference between the Pol and UnP groups still allowed us to address our hypothesis that slowing degradation would improve axonal regeneration across the injury gaps. This was supported by comparing the NF staining at the distal end of the conduits after 14 WPS to the parameters of degradation of filaments at 6 WPS. Both the SA and DR were significantly correlated with the NF score (Figure 8n–o). Both analyses omitted the animal shown in Figure 6e–f, and the NF score was missing tissues for one Pol animal, so the *n* values were 4, 5, and 5 for UnP, Pol, and PEO groups. While these are small numbers and subjective scales, the correlation supports our hypothesis that there was indeed some influence of filament intactness on nerve regeneration. As discussed above, the initial cells growing out of the nerve stumps may have attached and spread out better on the smoother Pol and PEO surfaces vs. the UnP, but a slower degradation rate in vivo of the Pol group might also have provided better cellular guidance across the gaps. With the PEO group, we speculate that the nature of the anodization layer led with most filaments to more significant pitting degradation. In addition to preventing cellular guidance, the rapid degradation might have displaced cells or caused toxicity due to greater production of hydrogen gas bubbles, higher Mg ion levels, or higher local pH values. In the study of MAO-oxidized Mg alloy bone pins, the higher degradation rates of MAO-anodized Mg after 3–4 weeks in vivo were shown to be accompanied by greater hydrogen gas bubbling around the pins [32]. However, in their study, they showed that the initial slowing of degradation vs. uncoated pins was favorable for bone growth. In our study, we did not see any advantage of PEO anodization, but we do show that slowing degradation rates (Pol vs. UnP) influenced nerve regeneration. Caveats are, as mentioned above, that our number of animals was small and the differences were also small, so further research is needed. We have also not explored the electroconductive properties of Mg metal, which could influence nerve regeneration, except to note that Mg is significantly more conductive than Ti (unpublished observations), which was our control for metal effects in general [52].

## 5. Conclusions

Anodization via PEO slowed degradation of pure Mg in vitro, but not in vivo, while polishing the Mg filaments slowed degradation through 6 weeks after implantation in vivo compared to unpolished filaments. The relative SAs and DRs of the filaments in vivo after 6 weeks were significantly correlated with the amount of NF (axons) quantified just distal to the conduit repair after animals were euthanized at 14 WPS. Our findings include the following: (1) Polishing as-received Mg filaments removed surface deposits and resulted in a smoother surface. (2) PEO anodization of Pol filaments, using a non-toxic salt solution, produced a thin oxidized layer that slowed in vitro degradation of pure Mg filaments after immersion in a physiological salt solution. (3) PEO anodization improved attachment of cells in culture compared to Pol filaments. (4) Mg filaments inside nerve conduits were well tolerated when used to repair a gap of 15 mm in rat sciatic nerves. (5) In vivo, PEO anodization did not slow degradation rates to 6 WPS, but introduced greater variability, while polishing Mg filaments slowed degradation compared to UnP filaments. (6) Function was not restored by any nerve repair with conduits, with or without Mg or Ti filaments, while isograft repairs restored partial functionality. (7) Small numbers of axons (NF-positive fibers) were found distal to the conduits in the Em and all three Mg filament groups. (8) The relative amounts of axons were positively correlated with the surface area of the filaments detected by micro-CT imaging at 6 WPS and negatively correlated with the 6 WPS DRs. In summary, our results provide further evidence of the safety of pure Mg for nerve regeneration and, while function was not restored with Mg, our NF data demonstrated that slowing degradation rates of Mg improved one measure of nerve regeneration. Further work is now needed to identify means other than anodization to slow Mg degradation and test effects on nerve regeneration.

## Figures and Tables

**Figure 1 materials-16-01195-f001:**
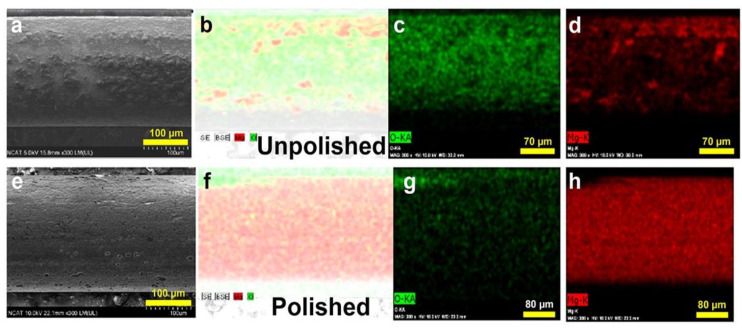
SEM and EDX analysis of Mg filaments as received (Unpolished) (**a**–**d**) and after acid treatment and polishing (Polished) (**e**–**h**). SEM images are shown in (**a**,**e**), merged EDX images in (**b**,**f**), oxygen in (**c**,**g**), and Mg in (**d**,**h**). With copyright permission from [35].

**Figure 2 materials-16-01195-f002:**
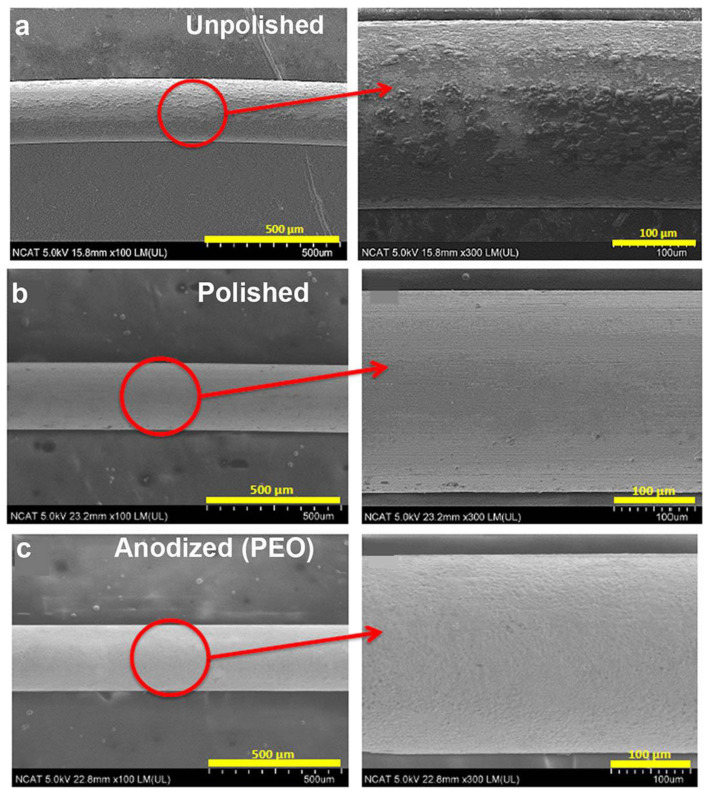
SEM imaging of (**a**) unpolished, (**b**) polished, and (**c**) PEO-anodized Mg filaments. With copyright permission from [35].

**Figure 3 materials-16-01195-f003:**
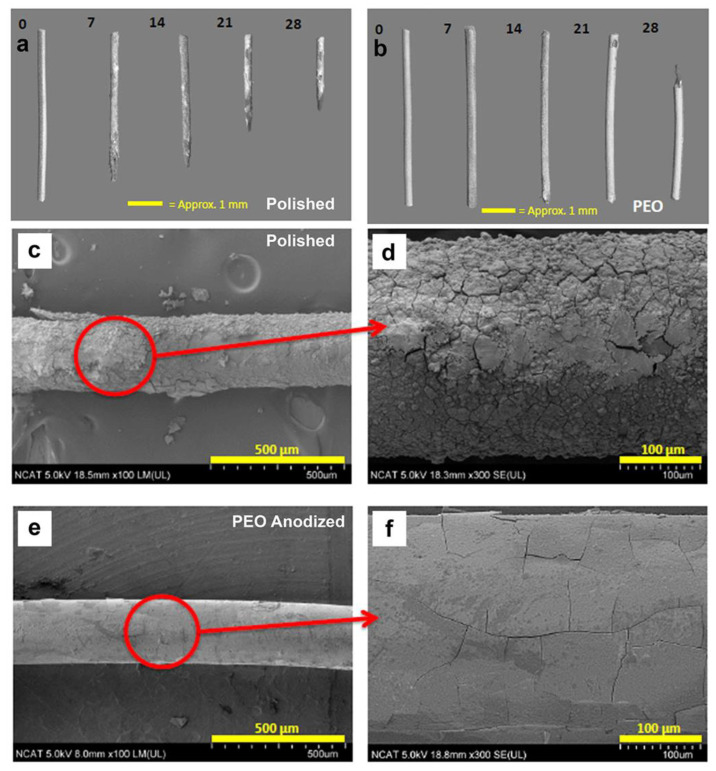
(**a**,**b**) Micro-CT images of Mg filaments before and at each week after suspension in HBSS for up to 4 weeks, showing 3D renderings. (**c**–**f**) SEM images of 4-week samples, showing surface morphology of a polished (**c**,**d**) and PEO-anodized (**e**,**f**) Mg filament at different magnifications. With copyright permission from [35].

**Figure 4 materials-16-01195-f004:**
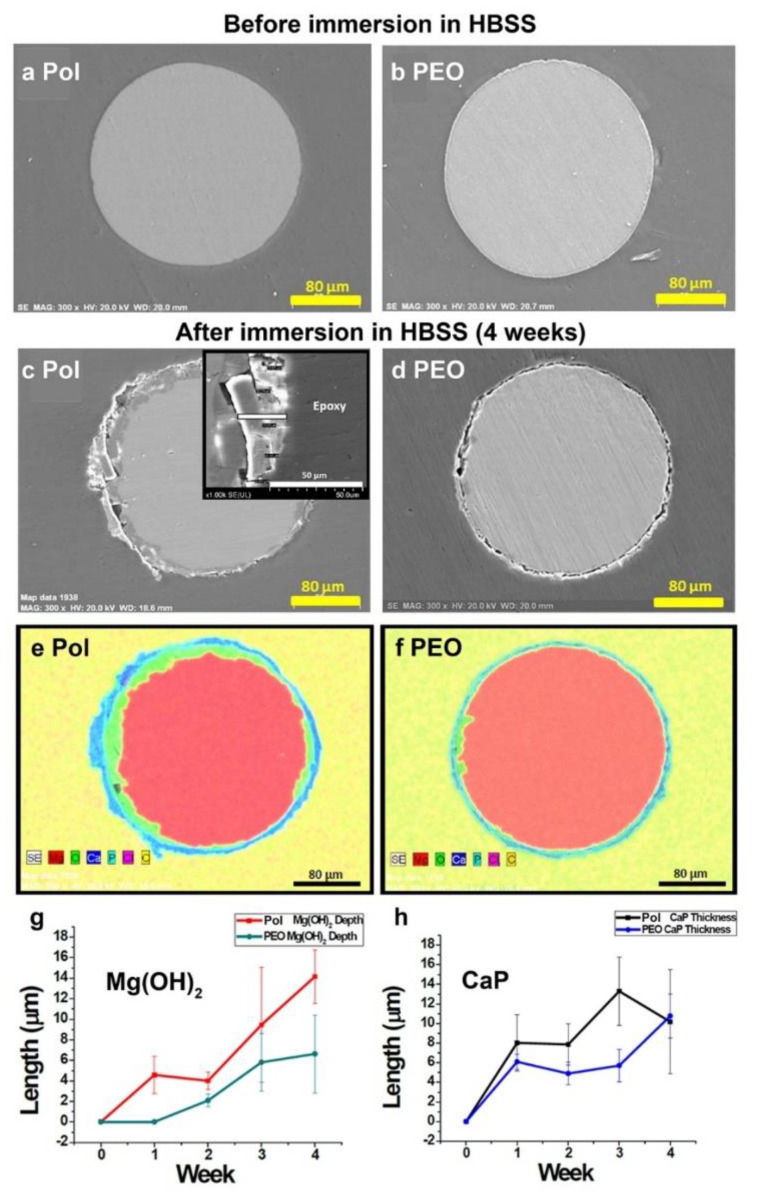
Effects of 4-week immersion in HBSS on Mg filaments through analysis of cross-sections. (**a**-**d**) SEM images of filaments before (**a**,**b**) and after (**c**,**d**) immersion. (**e**,**f**) EDX maps of filaments after immersion, merged element image. (**g**) Length (depth) of the Mg(OH)_2_ layer measured in EDX images. (**h**) Length of the calcium phosphate (CaP) layer. With copyright permission from [35].

**Figure 5 materials-16-01195-f005:**
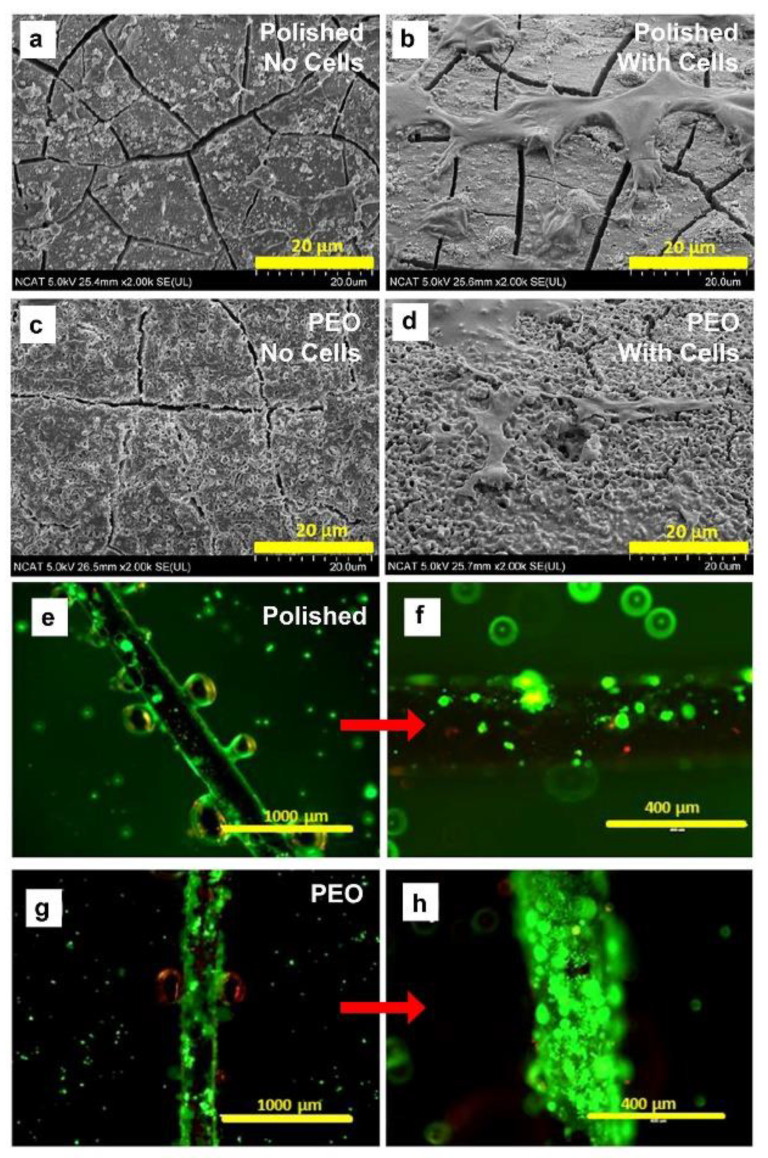
SEM imaging of the surface of Pol and PEO filaments after 12 h incubation in growth medium without (**a**,**c**) and with (**b**,**d**) PTBE cells added. A Live/Dead cell assay (live cells are green, red are dead) and light microscopy imaging showed that fewer viable cells attached to the Pol uncoated filaments (**e**) and higher magnification in (**f**) than to PEO coated filaments (**g** and higher magnification in (**h**). With copyright permission from [35].

**Figure 6 materials-16-01195-f006:**
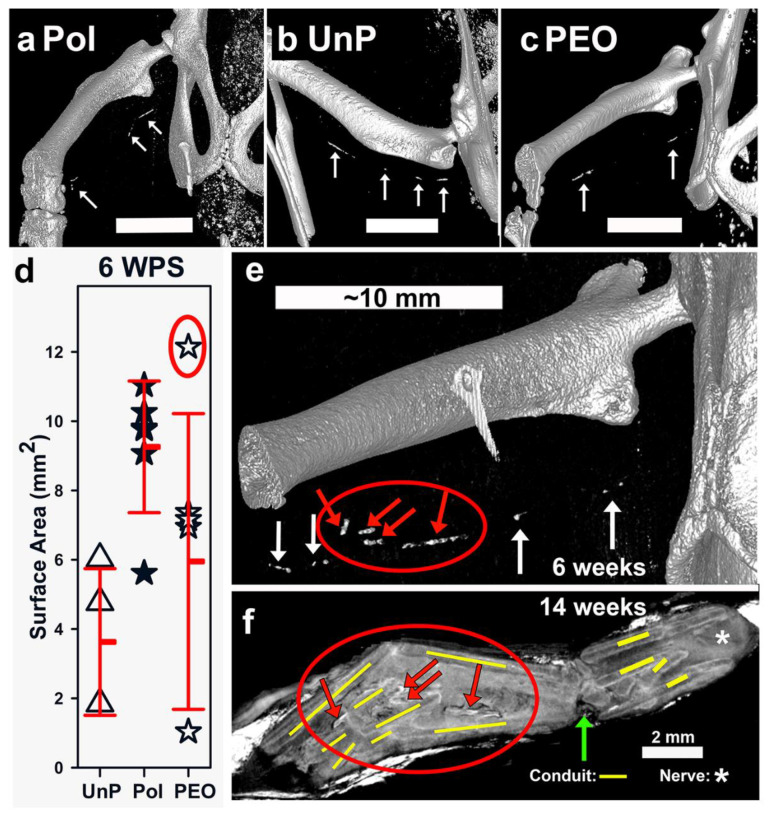
In vivo Mg filament degradation in nerve repair. (**a**–**c**) Live rat micro-CT 3D renderings at 6 weeks post-surgery (WPS), bars = 10 mm. (**d**) Live SA quantified at 6 WPS. (**e**) Live 6-week CT image of the rat with the largest SA, (red circle in (**d**) and red arrows inside the red circle show filament fragments off midline and semi-adjacent). (**f**) Micro-CT image at 14 WPS of tissues from same rat, with iodine contrast, showing degradation material in similar positions. Yellow bars illustrate broken conduit pieces. Asterisk (*****) shows proximal nerve tissue and green upward arrow shows a significant break in conduit and tissues.

**Figure 7 materials-16-01195-f007:**
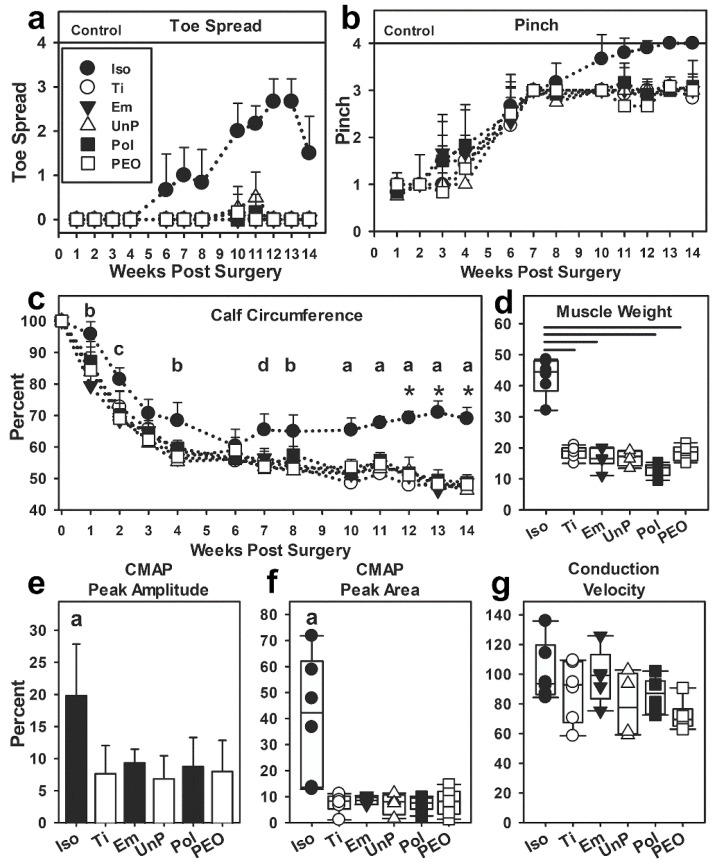
Functional measures. (**a**) Reflex toe spread in live animals. (**b**) Pinch test in live animals. (**c**) Muscle atrophy in live animals analyzed by measuring calf circumference (injured normalized to control leg as a percent, week 0 values are a place keeper, not used in statistics). Statistics indicated by letters are as follows: a = Iso > all groups; b = Iso > all but Pol; c = Iso > all but Ti; d = Iso > all but Em. The (*****) indicates Iso > its 6 WPS value. (**d**) After euthanasia at 14 WPS, dissected calf muscles were weighed. Bars indicate significant differences by SAS. (**e**–**g**) Electrophysiological measurements of nerve integrity were recorded immediately before euthanasia (injured/control leg as a percent). (**e**) Compound muscle action potential (CMAP) peak amplitude. (**f**) CMAP peak area. (**g**) Conduction velocity. Symbols are in Toe Spread legend.

**Figure 8 materials-16-01195-f008:**
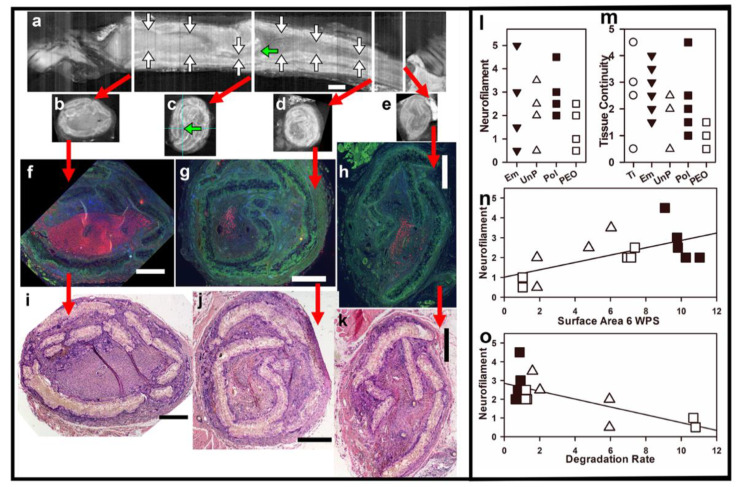
(**a**–**k**) Comparison between micro-CT and tissue histology from one Pol group animal. In Micro-CT images (**a**–**e**), conduit pieces look like double white lines and tissues have varied shades of gray (water is black). Breaks can be seen in the conduit and no Mg metal was detected. (**a**) In a longitudinal view, a continuous pathway (tissues between white vertical arrows) is seen from proximal (left) to distal ends. (**b**–**e**) Axial images were taken at the white lines shown in (**a**), as indicated by red arrows. Horizontal green arrows in (**a**,**c**) point to one of the narrowest tissue strands (blue grid lines cross in 8c on the tissue strand indicated by the green arrow). Bars are 1 mm for (**a**–**e**). In (**f**–**h**), paraffin sections matched to the micro-CT slices were immunostained for axons (anti-NF, red) and nuclei (blue) (green non-specific staining is shown to aid visualization of tissues). In (**i**–**k**), adjacent sections were stained with H&E. Magnification bars in (**f**–**k**) are 0.5 mm. Graphs (**l**–**o**): In (**l**) NF scores per group, (**m**) tissue continuity per group. (**n**) NF scores calculated at 14 WPS correlated with SAs at 6 WPS. (**o**) NF scores correlated with DRs at 6 WPS. Symbols are in Figure 7.

**Table 1 materials-16-01195-t001:** Effects of polishing on the atomic percentages of oxygen (O) and magnesium (Mg) on the surface, determined by EDX. With copyright permission from [35].

**Sample**	**Atomic %** **Mg**	**Atomic %** **O**
Unpolished	9.94	38.32
Polished	41.33	7.06

## Data Availability

Not applicable.

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
