# Peer review of "Effects of Altering Magnesium Metal Surfaces on Degradation In Vitro and In Vivo during Peripheral Nerve Regeneration"

_materials, 2023, doi:10.3390/ma16031195_

Round 1

Reviewer 1 Report

This paper is quite interesting and well suitable for the modern bio sciences. The Mg-biodegradable nature certainly encourages the modern thinking and it is very good case study. However, the text given in the results can be reduced, because in the discussion part it is covered all the aspects of prior to clinical as well post clinical test results. 

Some of the typographical errors observed in the paper are: 

1. Page-4, line 152, should be mm2 and mm3.. Similarly, line 187, mm2

2. page 6, line 267 is not clear. p-5, 239 line, make sentence simple. 

3 page 18,' image J; 

4 Page 17, figure captions are types in next page. like that 

3

Reviewer 2 Report

This is a study about effects of altering Mg Surfaces on degradation rate in vitro and in vivo during peripheral nerve regeneration. I suggest some revisions according to the following points:

Major points

1.      Why is there no sham-operated group to exclude muscle damage from surgery?

2.      In the EDX Images of Pol filament degradation in vitro after 4 weeks, why the content of Ca, P would decrease?

3.      What’s the purpose of Figure 4 (h)? If it is necessary to extend the observation period to judge the time of the peeling/flaking away of some of the Ca/P layer in PEO group? Please think over.

4.      It is recommended that the cell attachments and surface roughness of Ti be given to provide a more detailed description and comparison.

5.      In Figure 5 (d), there is also a data in Pol group that is outside the statistical range. What are the possible reasons for this result?

6.      Figure 5 and Figure 8 both show the nerve conduits fracture. Does the rupture of the nerve conduits affect the neurological function? And PCL rupture to a large extent in vivo, and the rupture time could not be determined. Will this have negative effects on neurological function?

7.      In Figure 8 (i), Why is the Ti group not shown?

8.      In Figure S4 (d), there seems no obvious linear relationship between degradation rate and S/V.

9.      The authors mentioned that except for the positive (Iso) group, there was no significant difference in the effect on motor function, yet the degradation rate did vary considerably in each Mg group. Mentioned in the conclusion of “support the concept that slowing degradation rates of Mg will improve nerve regeneration.” this view seems not consistent with the experimental results.

Minor points

1.      Incorrect number of figures in the article. (Line424 464 483 536)

2.      Line 308 310 314: the experimental images and the supporting material " Supplementary Figure S1 "does not match, it should be changed to" Supplementary Figure S5".

3.      Line 484: Notes of figure 5 (corrected) is wrong, “growth medium without (a and c) and with (b and c) PTBE cells added” should be corrected as “growth medium without (a and c) and with (b and d) PTBE cells added.”

4.      Line 561: “Figure S5” should be changed to “Figure S4”.

5.      Line 591: “S4” should be corrected as “S5.”

6.      Line 602: Please give the correct figure indication. (See Figure S5 for measurement method and Figure 6c, injured/control leg 602 circumference given as a percent).

7.      Notes of Figure S3 is incorrect. It should be the PEO group.

8.      Line 661: In Figure 8 (m), the ordinate is not DRs. Please reconsider it.

9.      Line 596: Please insert specific reference instead of showing “(ref)”.

Round 2

Reviewer 2 Report

The revised manuscript basically meets the requirements and can be accepted.